# Facial femininity and perceptions of eating disorders: A reverse-correlation study

**Valerie Douglas[1], Benjamin Balas[1]\*, Kathryn Gordon[2]**

**1** Department of Psychology, North Dakota State University, Fargo, ND, United States of America, **2** Sanford Research, Fargo, ND, United States of America

\* Benjamin.balas@ndsu.edu

**Data Availability Statement:** All data and materials used for the research described here are publicly available via the Open Science Framework (https://osf.io/kbmz7/).

## Abstract

Eating disorders are prevalent in college students but college students are not accurate in identifying the presence of eating disorders (ED) especially when race is involved. Much has been researched about diagnostic ability in vignette form, but little outside of this. For example, it is not known how facial features, such as perceived femininity, may affect observers' beliefs about the likelihood of disordered eating depending on race. In the present study, we examined how biases regarding facial appearance and disordered eating may differ depending on the race of face images. Using a technique called reverse correlation, we estimated the image templates associated with perceived likelihood of disordered eating using both White and Black Faces. Specifically, we recruited 28 college students who categorized White and Black faces according to perceived likelihood of an eating disorder diagnosis in the presence of image noise. Subsequently, we asked Amazon Mechanical Turk participants to categorize the resulting race-specific face templates according to perceived ED likelihood and femininity. The templates corresponding to a high likelihood of an ED diagnosis were distinguished from low-likelihood images by this second independent participant sample at above-chance levels. For Black faces, the templates corresponding to a high likelihood of an ED diagnosis were also selected as more feminine than low-likelihood templates at an above-chance level, whereas there was no such effect found for White faces. These results suggest that stereotyped beliefs about both femininity and the likelihood of disordered eating may interact with perceptual processes.

## Introduction

There are three main eating disorder (ED) diagnoses: anorexia nervosa, bulimia nervosa, and binge eating disorder [1]. Subclinical ED symptoms that do not fully cross the clinical threshold for a disorder can also be problematic [2] One particularly vulnerable group for the development of disordered eating symptoms is college-aged individuals, with an increased rate from 1995 to 2008 [3]. This is troubling considering that college students do not reliably recognize the symptoms of EDs, the need for treatment when presented with ED vignettes [4], and do not consistently identify what constitutes disordered eating [5].

**Funding:** This research was supported by NIH (National Institute of Health) grant NIGMS P30 GM1148748. The funders had no role in study design, data collection and analysis, decision to publish, or preparation of the manuscript. One of the authors (Balas) received summer salary from the grant listed in response to part (a) above.

**Competing interests:** The authors have declared that no competing interests exist.

This ability for college students to recognize EDs or the seriousness of EDs is further complicated by interference from stereotypes of EDs such as racial stereotypes [4, 6]. Besides affecting untrained college students, research has also found that clinicians tend to underestimate the severity of ED symptoms among African-American girls [7]. College students, who have not had training on diversity in mental illness, may be prone to a racial bias as well [6], although this effect was not replicated in a more recent study [4]. One aspect of racial stereotypes that has not been investigated in regard to EDs is how someone's facial appearance may influence the perceived likelihood of an ED separate from a vignette. That is, what beliefs do people have about what someone with ED looks like? Do these beliefs differ by race? While it may seem odd to speculate on stereotyped beliefs that observers may have about facial appearance and eating disorders, there is a large (and growing) literature describing how individuals make judgments regarding the personality characteristics and expected behavior of others based on facial appearance. Naïve observers make consistent judgments about what characteristics make faces look more trustworthy or dominant [8], more competent in leadership roles [9], and make consistent judgments regarding the mental health of unfamiliar individuals [10]. Many of these judgments are not valid: Perceived trustworthiness from face images does not reflect the likelihood that an individual will behave in a trustworthy fashion, for example [11]. For our purposes, however, it is observers' beliefs about correlations between facial appearance and the likelihood of disordered eating that are of interest: What biases do people have about the appearance of people with eating disorders and do these biases vary by the race of the people under consideration? Given that EDs are more likely to be diagnosed in females than in males [1], we hypothesized that observers would believe that faces that looked more feminine would also be more likely to have an ED diagnosis. Further, we hypothesized that this tendency might be affected by the race of the face under consideration due to additional observer biases affecting how femininity is perceived across racial categories.

We chose to test these hypotheses using a technique called reverse correlation. Reverse correlation (RC) is a powerful technique for revealing the internal representations that support perceptual judgments. Originally applied to low-level psychophysical research [12, 13], Reverse correlation designs have since been widely used to characterize the internal templates used in face categorization tasks. All image-based reverse correlation experiments share some common properties: Observers are typically presented with a stimulus image (sometimes called the *base image*) multiple times with some form of noise applied to it on each presentation so that each presentation is unique in terms of the combination of the stable original image and a noise pattern that is generated randomly on each trial. Examples of noise that may be added to the original stimulus images include the addition of white noise to each pixel in the image, or the application of "Bubbles" [14], which appear as randomly placed windows in a field of black pixels through which the underlying base image is visible. Presented with the combination of a base image and randomly generated noise, observers categorize each stimulus: Labeling the emotion expressed by a face, for example. These responses are used to analyze the noise patterns applied during the experiment according to how they were categorized by participants. Noise patterns that led observers to assign images to a particular category ("Happy") are grouped together and separated from noise patterns that led observers to assign images to different categories ("Sad"). By averaging together all of the noise patterns that led to a particular judgment (e.g. "Happy"), the experimenter can obtain an image that possibly reflects how some properties of the noise correlated with perceived membership in that category. That is, on average, what noise do we have to add to a neutral face so that it looks happy, or like a particular person [15]? Applying that aggregate noise pattern to the base image used in the experiment results in a *classification image* (or CI), which reflects the average stimulus appearance

that led to each category judgment. This classification image can be interpreted as a sort of perceptual template or internal prototype of the category under investigation.

This technique can reveal what parts of a face support different tasks like gender categorization or identification [16], or it can be used to examine other aspects of face categorization. For example, by varying the base images to which noise is applied (e.g. a familiar face vs. a stranger), we can determine how the visual information used to make categorization judgments may vary from stimulus to stimulus [17]. Similarly, we may also vary properties of the observer to see how variation across participants changes the visual information used for categorization judgments. Dotsch et al. [18], for example, found that prejudice towards Moroccan individuals affected the classification image that resulted from a race categorization task such that the Moroccan classification image looked angrier when generated by individuals with higher levels of prejudice. Whether or not these classification images truly reflect perceptual mechanisms supporting these inferences or instead illustrate how both perceptual and cognitive mechanisms contribute to the criteria used to make decisions about face images is challenging to determine unequivocally. Presently, we treat these images as a useful means of demonstrating what aspects of appearance are linked to categorization judgments as a way of examining how face race may affect the expectations or stereotypes naïve observers have about the appearance of individuals with eating disorders.

An important point to acknowledge is that reverse-correlation as it is being applied here does not establish any link between physiognomy and disordered eating behavior. That is, the templates we obtain from this technique do not reflect real correlations between facial appearance and behavior, personality characteristics, or other individual differences. A classification image estimate of an observer's "Happy" face template does not imply that all happy faces actually do look this way, for example, but instead reflects the template that this observer used to categorize faces as such. While one could conceivably attempt to measure correlations between facial appearance and disordered eating in a large sample of real individuals, such physiognomic research is ethically questionable (see [19] for a discussion of this point) and can reinforce stereotyping and prejudicial beliefs about what individuals with specific disorders or behaviors look like. By contrast, our use of reverse correlation in the current study is designed to reveal what biases naïve observers may have with regard to disordered eating. We do not endorse such biases as valid. Instead, we argue that identifying and characterizing such biases is an important step towards understanding how real individuals with and without disordered eating may face differential treatment (clinical and otherwise) based on race-specific biases that observers maintain about facial appearance and behavior. For example, if such biases indeed affect the way observers estimate the likelihood that an individual has an eating disorder, this could lead individuals with disordered eating to be discouraged from seeking treatment either by family, friends, or health-care professionals. To the extent that such biases interact with perceived race, this may in turn increase disparities in referrals and treatment for individuals with disordered eating. We applied this technique to determine how race category (White vs. Black) affected the templates used to assess the likelihood of an eating disorder diagnosis based on facial appearance. In particular, we examined whether or not perceived femininity was differentially linked to perceived likelihood of having an eating disorder diagnosis as a function of race.

## Experiment 1 – Do faces that look likely to have an eating disorder look more feminine to naïve observers?

In our first experiment, we recruited two samples of participants: (1) A group of observers who completed a reverse correlation task using images of White and Black female faces. (2) A

group of workers recruited via the Amazon Mechanical Turk who provided forced-choice judgments using the classification images created by the first group of observers.

## Methods

**Participants.** We recruited 28 participants (13 female) between the ages of 18–23 to take part in this experiment. All participants were recruited from the North Dakota State University (NDSU) Undergraduate Psychology Study pool and received course credit for volunteering. All participants were naïve to the purpose of the experiment, reported having either normal or corrected-to-normal vision, and gave written informed consent prior to the experimental session. Given our interest in understanding the perceptual biases maintained by the general public with regard to race and disordered eating, we did not include individuals in this sample who had clinical experience with individuals with an eating disorder diagnosis. This is an interesting question in its own right, as clinicians may maintain their own set of different perceptual biases regarding facial appearance and disordered eating behavior that would be important to understand in the context of practitioner-patient relationships. At present, however, we opted to examine perceptual biases in an inexperienced sample to more broadly characterize what perceptual templates naïve observers may use to make judgments about disordered eating. Finally, we note that our sample was predominantly White individuals (26 White participants, 2 Black participants), which limits our ability to comment on how observer race may affect our results. Due to the small number of Black participants, we do not include observer race as a covariate in any of our analyses and refrain here from offering comparisons between the responses of White participants and Black participants due to the low statistical power for doing so. As such, our results should be interpreted primarily as reflecting the biases of White observers, with unknown generality of these results as a function of observer race.

**Stimuli.** We used two 400x400 pixel grayscale "Average" female faces (White and Black) as base images for our task (Fig 1). These images were created using commercial software (Face Gen, Singular Inversions) that includes the capability to manipulate 3D models of faces of multiple races, including an average face for each race.

Because we used FaceGen faces as stimuli in both experiments reported here, it is important to discuss how the underlying stimuli (average faces for race and gender categories) were created within the software. Briefly, the FaceGen model is based on high-resolution 3D scans of ~270 real human faces including men and women from varying racial categories. These scans include both 3D shape information describing the geometry of the head and face as well as 2D albedo information describing how pigmentation varies across the surface of the skin. These measurements have been used to develop parametric models of face shape and face pigmentation (a "face space") that account for the variance in those head models. That is, given this particular sample of faces described in terms of their 3D shape and 2D surface properties, this compact dimensional model of the data based on the faces makes it possible to assign a small number of spatial coordinates to each face such that faces can be meaningfully characterized in terms of their distribution within this space. Random FaceGen faces can then be sampled by gender, race, or other categories by randomly sampling from the distributions that correspond to those categories. Average faces (such as the faces described above) can be sampled from each category by identifying the point that lies at the center of the distribution of faces belonging to that category. The images created by this procedure thus are limited by the characteristics of that sample, but nonetheless are based on real people and their appearance variability. As such, these average faces reflect differences in shape and surface pigmentation that correspond to category differences in face appearance as reported by the individuals who contributed to the scanned dataset. While this procedure certainly is limited in scope by the nature of

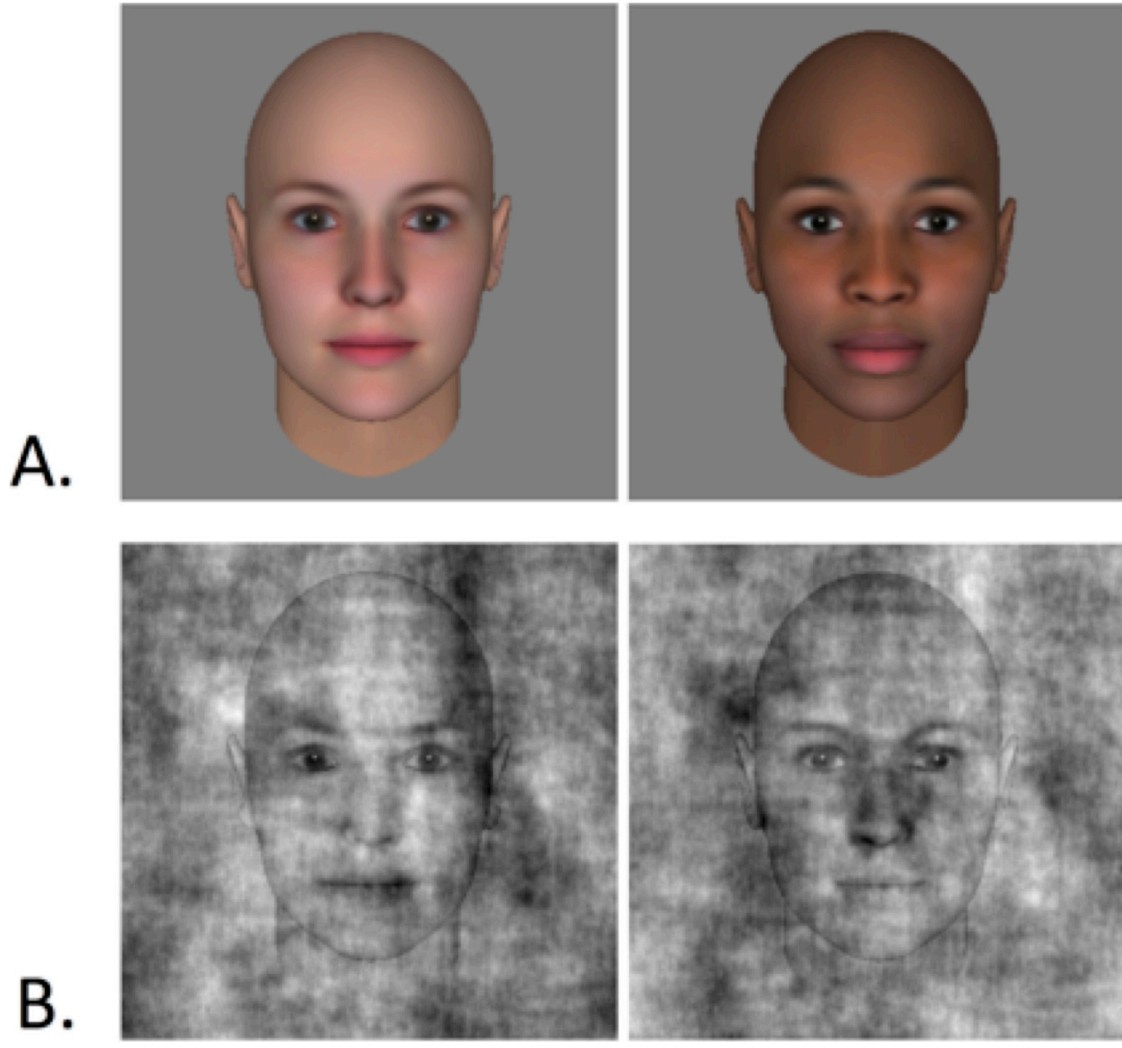

**Fig 1. Examples of the stimuli used in our experiment.** (A) The average White face and average Black face used as the basis for all trials. (B) An example of the average White face with noise added–the two noise patterns are inverted versions of each other, giving rise to noticeable appearance differences in the two images despite the use of the same base image in each case.

the training corpus, we suggest that these capture sufficiently meaningful and representative appearance differences to be useful for our purposes.

**Procedure.** Participants completed a 2AFC categorization judgment using the images described above. On each trial, participants were presented with either two versions of the "Average White" image or two versions of the "Average Black" image, presented to the left and right of the center of the screen. Critically, these two images were created by adding two different noise patterns to the target face, resulting in appearance differences between the two stimuli. Participants indicated on each trial which face looked more likely to depict someone who had an eating disorder diagnosis. We chose to ask this fairly broad question about eating disorder likelihood rather than asking participants to categorize faces according to a more specific diagnosis (e.g. anorexia nervosa or bulimia) for two reasons. First, given that our sample was comprised of naïve observers, we anticipated that many of our participants might not be familiar with these more specific terms and we wished to avoid providing descriptive information prior to completing the task for fear of influencing the criteria they might use to respond to our

stimuli. Second, we felt that phrasing the question in broad terms might encourage participants to think more generally about the mental health of the faces they were presented with, which we think is potentially more representative of a layperson's understanding of the topic.

These noise patterns were created online using custom routines in Matlab–one pattern was made by generating random 1/f fractal noise, and the second image was made by "negating" the first: Light pixels in one image corresponded to dark pixels in the other, and vice-versa. This maximizes the potential for the two images presented on each trial to look different from one another (Fig 1B). Participants used the "z" and "m" keys on the keyboard to choose the left or right face, and both their reaction time and the noise patterns applied to the target face on each trial were recorded.

Participants completed 500 trials in a single session (250 trials per race), with race category pseudo-randomized across trials for each participant. We presented the task on a 1200x800 pixel LCD monitor, and all stimulus display and response collection routines were controlled by custom software written using the Matlab Psychophysics toolbox [20, 21].

All procedures described above (and in Experiment 2) were approved by the North Dakota State University Institutional Review Board (#SM11168). Written informed consent was obtained from all participants.

## Results

We computed each participant's CI corresponding to the average appearance of White faces and Black faces that were selected as looking more likely to have an eating disorder diagnosis (we will call these "+ED" images). We calculated these by averaging together the noise patterns added to the faces participants selected on each trial and adding that average noise pattern to the original target face. We also carried out the same process for the unselected images, yielding the average appearance of White and Black faces that looked *less* likely ("-ED" images) to have an eating disorder diagnosis (Fig 2).

In some reverse correlation studies, researchers determine which regions in the image significantly correlate with a categorization decision [22]. Our goal, however, was to determine how classification images varied by race category according to their apparent femininity. To do so, we presented all of the individual classification images generated by our participants to a new group of online observers via the Mechanical Turk. Specifically, we paired each +ED image with its corresponding–ED image and asked separate groups of online observers to choose the face that looked: (1) More likely to have an eating disorder, and (2) More feminine. The first judgment serves a purely confirmatory purpose: Do the classification images created by our observers based on their perceptual templates for disordered eating look more likely to have an eating disorder diagnosis to independent raters? The second judgment addresses our critical question: How does femininity co-vary with image features related to the perception of disordered eating, and does this relationship vary by race? Again, we emphasize that these relationships do not reflect valid correlations between facial appearance and behavior, but instead reflect the biases maintained by our observers. We also note that we did not collect demographic information about the participants who completed this task using the AMT platform. We expect that this sample of participants differs from the predominantly White college students who completed our first task, but beyond that we cannot make strong statements about the demographics of this group. As such, we again do not make any attempt to include observer race, age, or nationality as a covariate in our analyses, opting instead to simply report the aggregate statistics for the 2AFC judgments used here. In terms of the generality of our results, above-chance classification of the templates generated by our college-student sample by an independent sample of participants recruited from the much broader population of

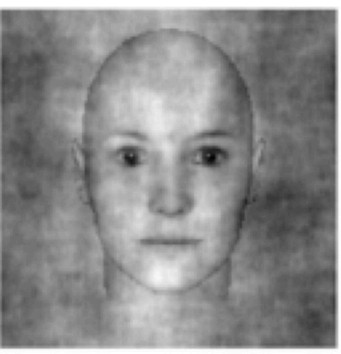
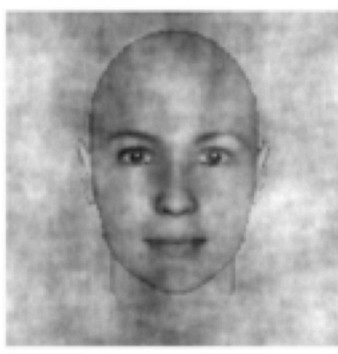

White Faces (+ED)  White Faces (-ED)

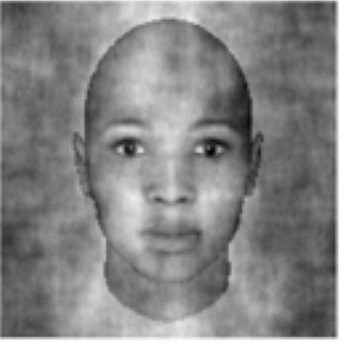
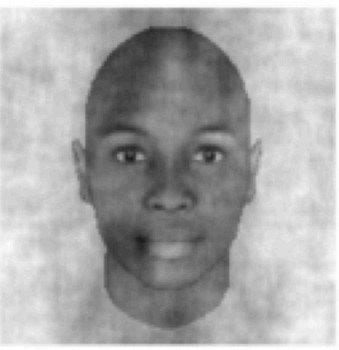

Black Faces (+ED)  Black Faces (-ED)

**Fig 2. Average classification images for White (top row) and Black (bottom row) faces.** These represent the average perceptual template for faces that looked likely to have an eating disorder diagnosis (+ED) and those that looked less likely to have such a diagnosis (-ED), calculated by averaging together the individual templates estimated for each observer.

AMT workers to some extent demonstrates that the perceptual judgments made by our more specific sample are fairly robust. At the very least, the different populations contributing to these tasks do not offer an easy account of the data that invalidates our main conclusions.

We collected a total of 560 independent judgments for each question for both White faces and Black faces. While we cannot conclusively rule out the possibility that some AMT participants may have used multiple accounts to make many responses, we treat these judgments as independent categorizations made to each stimulus by individual AMT workers. For both White faces and Black faces, the "+ED" face was chosen as looking more likely to have an eating disorder at above-chance levels (White faces, 56%, $p = 0.003$; Black faces, 58%, $p < 0.001$; two-tailed binomial test). The proportion of "+ED" faces selected did not differ as a function of race (Chi-squared = 0.23, $df = 1$, $p = 0.63$). In terms of femininity, however, we observed different outcomes as a function of race. For Black faces, the "+ED" face was significantly more likely to be chosen as the more feminine looking face (66%, $p < 0.001$, two-tailed binomial test), but we observed no significant preference for White faces (49.9%, $p = 0.97$, two-tailed binomial test). Critically, these two proportions did differ as a function of race (Chi-squared = 14.48, $df = 1$, $p < 0.001$). All of these analyses were carried out using the JASP statistics application [23].

The classification images created by in-lab participants were thus reliably interpreted by independent observers as looking more likely to have an eating disorder across both White and Black faces, but race affected how perceived femininity covaried with the appearance characteristics associated with disordered eating by our first group of observers. In Experiment 2, we further examined this interaction between race, perceived femininity, and perceived likelihood of eating disorder by directly manipulating the appearance of White and Black faces using the FaceGen model's estimates of facial femininity.

**Image statistics analysis.**   One concern regarding the perception of the images generated by the reverse correlation is that both the foreground and the background of the images created for Experiment 1 were subject to the addition of noise. This means that while we assume that differences in how the resulting face images are perceived by independent observers result from varying facial appearance, it is possible that characteristics of the background could systematically bias perception as well. This is especially relevant to our judgments of femininity. Female faces differ from male faces in terms of skin luminance and higher contrast between facial features [24], and an image background that is systematically brighter or darker could induce a change in the perceived lightness or contrast of the face pattern through simultaneous contrast effects. To determine if this could be the case in the reverse correlation images our participants generated, we analyzed a set of image statistics extracted from the foreground (face) and background of these images.

To segment foreground pixels from background pixels in each reverse correlation image, we created binary masks for the original White and Black faces used in Experiment 1. We created these masks by selecting the entire background in each original image, inverting that selection so that the entire face was selected instead, and then deleting the face pattern from the image. This yielded images with zero values where the face pattern had been and uniform intensity values everywhere else. Next, we imported these masks into MATLAB along with the reverse correlation images. The masks were used to select either the foreground or background pixels of each image for further analysis. We chose to calculate the mean intensity of the foreground and background of each image, which would allow us to determine if there were systematic differences in how light or dark the background of the reverse correlation images were across categories. Specifically, we characterized the foreground and background luminance by calculating the average pixel intensity (grayscale value) across all pixels in each region. This yields a single foreground and background value per image, which can then be compared across conditions to look for systematic differences.

We analyzed the mean luminance values from the foreground and background separately by testing the hypothesis that the difference between these values for ED+ and ED- images was non-zero. Because these values were constrained to be redundant by our use of positive and negative noise patterns on each trial, we divided the difference between ED+ and ED- values in half before using a single-sample $t$-test to compare the resulting difference scores to zero. This procedure was carried out using the JASP statistics application [23]. This analysis revealed that neither White faces ($t$ (27) = -1.55, $p$ = 0.132) nor Black faces ($t$ (27) = -2.21, $p$ = 0.035) had systematically darker backgrounds for ED+ face images compared to ED- face images after correcting for multiple comparisons via the Bonferroni correction (reducing the critical alpha value to 0.025). It is evident that there is a trend favoring darker ED+ backgrounds for Black faces, but our analysis suggests that this is not an overwhelmingly large effect (Table 1).

The foregrounds of both White and Black face images, however, did differ in luminance across ED+ and ED- images. Specifically, both White faces ($t$ (27) = 3.56, $p$ = 0.001) and Black faces ($t$ (27) = 2.95, $p$ = 0.006) tended to have higher intensity values in ED+ images than ED- images (Table 2).

**Table 1. Descriptive statistics for background intensity analysis.**

| race | response | Mean | SD | N |
|------|----------|------|-----|---|
| White | ED+ | 128.6 | 25.85 | 28 |
| | ED- | 143.7 | 26.08 | 28 |
| Black | ED+ | 137.0 | 25.34 | 28 |
| | ED- | 156.1 | 21.67 | 28 |

This outcome demonstrates that there is a clear effect of image type on the intensity of image foregrounds, but not image backgrounds. Thus, the differences in how femininity and eating disorder likelihood are perceived in these images are primarily due to the appearance of the face itself and not induced by the surrounding region. There are of course a great many other image statistics one can examine besides mean intensity, but the relationship between background and foreground structure are less well understood. The effects of surround contrast on the perception of a central target are complex, for example, and depend on the relative orientation of image structure in the surround and the target (which is not controlled or manipulated here) and also on the relationship between the surround and target contrast [25]. We therefore have chosen to only include this relatively simple image analysis here, and have provided code for carrying out this analysis and the segmentation code to support more complex statistical extraction via the Open Science Framework (https://osf.io/kbmz7/).

## Experiment 2 – Do feminized faces look more likely to have an ED?

In our second experiment, we asked naïve observers to determine whether a novel face or a feminized version of that same face looked more likely to have an eating disorder diagnosis. Based on the outcome of Experiment 1, we hypothesized that feminized Black faces would be more likely to be selected as having an eating disorder than feminized White faces.

### Methods

**Participants.** In Experiment 2, we collected data from 400 Amazon Mechanical Turk workers. We elected to use this platform for this task primarily because we only had need of a small number of judgments in this experiment but wished to obtain these judgments from a large pool of participants. As described in Experiment 1, we did not collect demographic data regarding these participants and thus analyze our data only in the aggregate without covariate analyses of observer race or gender. Again, we anticipate that this participant sample is likely more heterogeneous than our original sample of predominantly White college students in Experiment 1, but beyond that do not make strong statements about the generality of the main results.

**Stimuli.** We used FaceGen to create novel full-color images of White and Black faces via random sampling of the FaceGen model space. Specifically, we drew samples that were constrained to be female, White or Black (according to the distribution of faces used to create the

**Table 2. Descriptive statistics for foreground intensity analysis.**

| race | response | Mean | SD | N |
|------|----------|------|-----|---|
| White | ED+ | 141.1 | 9.777 | 28 |
| | ED- | 128.3 | 10.825 | 28 |
| Black | ED+ | 122.6 | 16.566 | 28 |
| | ED- | 104.5 | 17.500 | 28 |

FaceGen database), and slightly more average-looking than FaceGen's parameters for "typical" appearance. We generated 10 images per race category using these parameters. Next, we manipulated the femininity of each of these faces using FaceGen's slider-bar interface for continuously varying the appearance of a target face, yielding a total of 40 images (10 original White Faces, 10 original Black faces, and the feminized counterpart of each). Again, this interface facilitates sampling from the dimensional model established by the original laser-scanned faces used to create the FaceGen software. Increasing the femininity of faces within the model corresponds to drawing samples that are further from the center of the distribution of male faces and closer to the distribution of female faces, which can be computed either across all races or with regard to a specific race category. In Fig 3, we display examples of each stimulus type. As we described previously, the parametric space underlying this interface is limited by the characteristics of the sample used to build the FaceGen appearance model. Nonetheless,

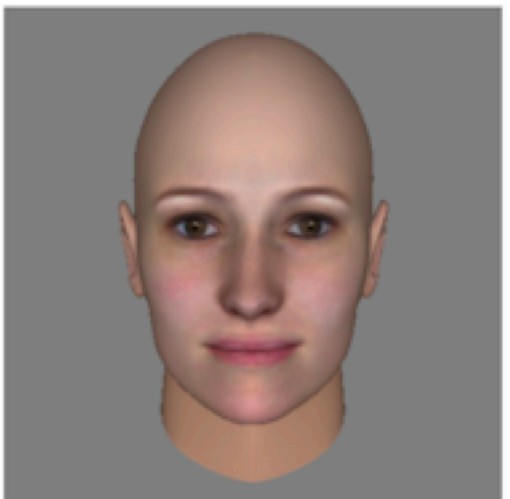 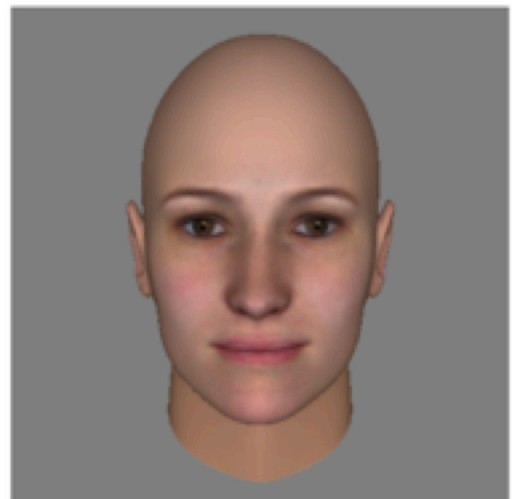

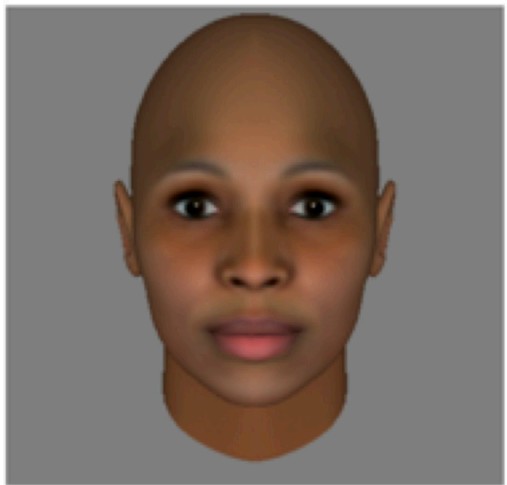 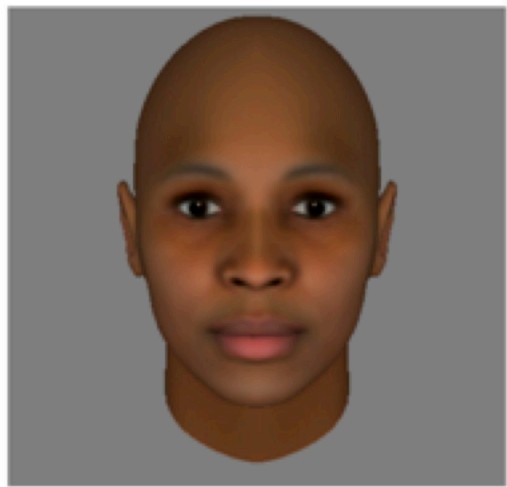

**Fig 3. Examples of original (right column) and feminized faces (left column) used in Experiment 2.** In each case, an original face was used as the basis for creating a feminized version of the same individual using FaceGen's model space.

varying faces according to FaceGen's sense of femininity does correspond to meaningful appearance variation across male and female faces both in terms of 3D shape and 2D surface properties. As in Experiment 2, we suggest that this makes it a useful choice for our purposes despite the constraints on its generality.

**Procedure.** We presented image pairs comprised of an original face and its feminized counterpart to Mechanical Turk workers, and asked workers to choose which of the two faces looked most likely to have an eating disorder diagnosis. Participants had unlimited time to make their judgment, and we collected a total of 20 judgments per image pair (a total of 400 judgments).

## Results

We found that for both White and Black faces, observers tended to choose feminized faces more than original faces as looking more likely to have an eating disorder diagnosis (White faces = 62.5%; Black faces = 71.5%, two-tailed binomial tests against chance each with $p < 0.01$). This difference reached marginal significance (Chi-squared = 3.65, $df = 1$, $p = 0.056$), suggesting that there is at most a weak effect of race on the outcome of this judgment.

## Discussion

The current study investigated the nature of observers' perceptual biases concerning the appearance of White and Black faces and the likelihood that individuals have an eating disorder diagnosis. Although there is vignette-based evidence that race affects the rates of being diagnosed with an eating disorder (e.g., [6, 8]), no research has investigated how one's facial appearance may affect the likelihood of being diagnosed with an eating disorder or being perceived as being likely to have disordered eating. Using reverse correlation methods, we determined sets of Black or White faces that were categorized according to the likelihood of having an eating disorder diagnosis or not and then separately confirmed these judgments with an independent group of observers. In addition, we assessed how perceived femininity of the faces affected the diagnosis of an eating disorder. We found that for Black faces, perceived femininity was associated with the perceived likelihood of having an eating disorder diagnosis. However, no such link was found for female White faces. In our second experiment, directly manipulating femininity to influence eating disorder categorization did not lead to differences in the outcome for Black and White faces, though we did observe a trend favoring a larger impact of feminization on the rate at which feminized Black faces were categorized as more likely to have an eating disorder diagnosis. Our data across the two experiments thus differ, potentially because our second experiment involved manipulations of feminine appearance that were not the same in magnitude as the differences in perceived femininity in ED+ and ED- templates. More careful work to examine the magnitude of femininity differences between faces perceived as more or less likely to have an eating disorder diagnosis may help provide a better sense of how these perceptual variables are related to one another.

Due to eating disorder having a higher prevalence rate for females than males by a factor of between 3:1 and 5:1 depending on the specific eating disorder [1], we predicted that it would be likely that, in general, femininity of the face would be a relevant characteristic for observers' judgments. However, we did not expect the importance of facial femininity in the determination of an eating disorder diagnosis to vary by race. Perhaps due to the stereotyped belief that White women are most likely to have an eating disorder, femininity may not be the most important marker of this stereotype among White faces. For Black faces, on the other hand, since their race violates the stereotype of who is more likely to have an eating disorder, assessors may tend to focus more on the relevant characteristic they have that goes with the

stereotype—being female. As being female becomes the characteristic of focus for Black faces, an increase in facial femininity leads to a stronger association with eating disorder. Critically, there is no evidence that this is at all a valid correlation that should be used to guide diagnosis. Instead, our work suggests that this may be a perceptual bias that affects judgments related to eating disorder diagnosis and should perhaps be taken into consideration when training care providers who make referrals regarding disordered eating or determine interventional strategies for individuals with eating or body image concerns. Future research should focus on assessing other relevant facial features that may be of stronger focus in White faces, as well as determine if facial femininity is also the most important characteristic in faces of other races. Facial attractiveness, perceived health based on facial appearance, and a wide range of other variables that observers estimate from face images (whether these are valid judgments or not) may also make contributions to how expert and naïve observers alike form first impressions of individuals based on facial appearance. In particular, given the role of skin lightness in our observers' classification images, our work suggests that further examination of how pigmentation cues that observers interpret in terms of physical health and attractiveness may be an important extension of this study. More complex descriptors of skin pigmentation are known to contribute to perceived age, health, and attractiveness [26, 27], and consideration of these features may help elucidate what critical information observers use to make judgments regarding the likelihood of disordered eating behavior in unfamiliar individuals. We also note that eating disorders occur in men as well as women, and applying our approach to an examination of perceptual biases that affect judgments of eating disorder likelihood in male faces could further establish how first impressions of variables like masculinity/femininity [28], attractiveness, and physical health may be further used to make biased inferences about other behaviors.

In general, these results indicate that facial appearance may affect decision-making related to the estimated likelihood of health-related behaviors in others, in this case, disordered eating. The existence of race-based perceptual biases that influence these judgments may lead to disparities in the identification and treatment of eating disorders. Minimizing the impact of these perceptual biases is an important direction for future work, as is continued investigation of other perceptual biases that may affect other category judgments that determine how physical and mental health needs are addressed.

## Acknowledgments

Thanks to Dan Gu and Ganesh Padmanabhan for technical support.

## Author Contributions

**Conceptualization:** Valerie Douglas, Benjamin Balas, Kathryn Gordon.

**Data curation:** Benjamin Balas.

**Software:** Benjamin Balas.

**Writing – original draft:** Valerie Douglas, Benjamin Balas, Kathryn Gordon.

**Writing – review & editing:** Valerie Douglas, Benjamin Balas, Kathryn Gordon.

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
