## [Decision Letter · Decision Letter 0]

26 Aug 2020

PONE-D-20-16005

Facial Femininity and Perceptions of Eating Disorders: A Reverse-Correlation Study

PLOS ONE

Dear Dr. Balas,

Thank you for submitting your manuscript to PLOS ONE. After careful consideration, we feel that it has merit but does not fully meet PLOS ONE’s publication criteria as it currently stands. Therefore, we invite you to submit a revised version of the manuscript that addresses the points raised during the review process.

May I first place on record my apologies for the delay in the review process. I have taken the decision to proceed with the comprehensive single review below, and will take on a second reviewer in the next round. I have invited an incredible amount of reviewers but in this time of civil upheaval, academics have rightly not been able to find the time over the summer to take on the review. Once again, my apologies. 

The work you present is very interesting and a novel analysis applied to facial femininity and the perception of eating disorders. As such, there is the potential to warrant publication in PLOS ONE. I hope your review can consider the points raised below and we can move forward. May I also ask that you consider making your data and analysis code/script publicly available in order for this work to become transparent and open. 

We look forward to receiving your revised manuscript.

Kind regards,

Darren Rhodes

Academic Editor

PLOS ONE

Journal Requirements:

"This study was approved by the NDSU IRB (#SM11168). Written informed consent was obtained from all participants.".

i) Please amend your current ethics statement to include the full name of the ethics committee/institutional review board(s) that approved your specific study.

ii) Once you have amended this/these statement(s) in the Methods section of the manuscript, please add the same text to the “Ethics Statement” field of the submission form (via “Edit Submission”).

"This research was supported in part by NIGMS P30 GM1148748, which supports the

Center for Visual and Cognitive Neuroscience at NDSU."

"The authors received no specific funding for this work."

Reviewers' comments:

Reviewer's Responses to Questions

**Comments to the Author**

1. Is the manuscript technically sound, and do the data support the conclusions?

Reviewer #1: Partly

2. Has the statistical analysis been performed appropriately and rigorously? 

Reviewer #1: N/A

3. Have the authors made all data underlying the findings in their manuscript fully available?

Reviewer #1: No

4. Is the manuscript presented in an intelligible fashion and written in standard English?

Reviewer #1: Yes

5. Review Comments to the Author

Reviewer #1: 1. Summary

The research report “Facial Femininity and Perception of Eating Disorders: A Reverse-Correlation Study” applied the approach of the Reverse Correlation (which I find novel and rare) to examine, whether some concrete facial cues are associated with the perception of Eating Disorders (ED). The research focused on the problem of raising ED prevalence in contemporary US College population. It used artificial stimuli from FaceGen database and two distinct groups of raters; US college students and Amazon Mechanical Turk (AMT) workers. Specifically, the research used female facial representations of two racial origins (faces with black and white complexion) and either set of faces with artificial white noise or set of faces manipulated in sexual dimorphism. Authors report a racially-specific association between sexual dimorphism (facial femininity) and the perception of EDs. When noise was not applied to the images, both black and white female facial representations with feminised features were significantly more frequently perceived as suffering from Eating Disorders. However, when noise was applied on non-distorted (non-feminised) female faces, only black female faces that were perceived as ED-positive were at the same time identified as more feminine. The noise patterns that caused non-feminised (unmanipulated) white female faces to be perceived as ED-positive were not significantly associated with noise patterns that made their appearance more feminine. This result was based on two independent datasets of raters (College students and AMT workers). The authors also present an analysis of the effect of an image background and foreground darkness and root-mean-scale contrast on the perception of ED and femininity. The perception might have been affected by the difference in contrast and foreground/background luminance. The authors conclude that the perception of eating disorders is stereotypically associated with femininity only in black female faces. In white female faces, perceived femininity is not a cue to an ED.

The authors have brought convincing arguments that EDs are an issue in current US society. They also cited references to the studies showing that both students and clinicians are inaccurate in diagnosing the EDs in female faces. This accuracy may be further affected by stereotypes based on the sex typicality and ethnicity of patients. People stereotypically think that women of European origin possess eating disorders more frequently. The authors utilised two independent sets of raters (college students and AMT workers). At above-chance levels, the AMT workers and college students agreed on the diagnosis of ED in the stimuli faces. The authors also showed an important difference between white and black faces concerning the association between ED and PERCEIVED femininity. The goal to determine how race category (White vs Black) and perceived/manipulated femininity affect the likelihood of being perceived as a patient with ED-diagnosis has been fulfilled.

2. Primary objections

Nonetheless, author should address the following objections:

(a) In my opinion, authors should explain the evolutionary relevance and adaptive consequences of the fast and accurate perception of cues of an eating disorder in a female face. Any perceiver should struggle to maximise his or her fitness. Accordingly, it is adaptive to assess women’s impaired health and fertility accurately. Eating disorders have been shown to multiply the risk of death and impair fertility [1,2]. Therefore, I would appreciate a review section on the effects of eating disorders on perceived health and attractiveness. Perceived health seems to predict actual health partially [3,4]. Perceived attractiveness and healthiness are also essential cues in mate choice, mate success, and status negotiation [5,6]. People should perceive markers of eating disorders as cues of impaired health and fertility, low social status and unwanted social and mating partner.

Moreover, the noise the authors applied on the faces can be interpreted as an alteration of the texture of facial skin, which seems crucial for the perception of female youthfulness and fertility [7,8]. I think a paragraph on evolutionary and mate-choice consequences of impaired health would highlight the crucial importance of improving the knowledge of perception of eating disorders. It would also "anchor" the research topic in the paradigm of evolutionary psychology.

(b) I do not understand why the authors use the method of adding a "noise" into artificial FaceGen faces to study “how does femininity co-vary with physiognomic cues related to the perception of disordered eating, and does this relationship vary by race?”. I would prefer to use natural faces of ED-positive and ED-negative individuals, get them rated on perceived femininity and presence of eating disorders. Then, I would analyse skin texture and facial shape to identify which cues in facial surface and shape make their appearance more ED-like. Following this, I would manipulate the faces to make them more ED-like and get those manipulated faces rated again. By the proposed attitude, I could identify cross-ethnical differences in various cues’ effect. If I ever wondered, which cues make people perceive someone ED-positive, I would use the methodology described above (texture and shape analysis, natural stimuli). And even if I used the method of "random noise", I would also try to identify, which patterns of noise drive the perception of ED-positivity and femininity! The face texture is of key importance for the perception of youthfulness, attractiveness and healthiness (see above). Nonetheless, authors do not identify which features of the texture drive the perception of ED and femininity. The most prominent "take-home message" is that "some pattern of noise is perceived as more feminine. Unlike in White women, in Black women, this noise is also perceived as associated with eating disorders". Still, the manuscript presents a good starting point concerning studying the perception of facial cues affecting the perception of Eating disorders. With regard to the PLOS ONE policy ("PLOS ONE uses peer review to determine whether a paper is technically rigorous and meets the scientific and ethical standards for inclusion in the published scientific record") the manuscript is in good order. However, authors apply a rare method, withouth further explaination (and justification). Authors might have been completely aware of the benefits of their method to a degree that these benefits are not mentioned (e.g. that it is easier than to get facial photos of people possessing ED-diagnosis). Nonetheless, such method might have been a result of a decision like "lets' use the RC method on the topic of EDs: the method of RC is novel and rare, the topic of ED seems important".

(c) The section “Image statistics analysis” (p. 11-14), is, in my opinion, excessive. The authors attempted to determine the effect of differences of background and foreground darkness and also the effect of root-mean-square contrast of both background and foreground on the perceived attributes. The section seems to me to be based upon request or advice of other (previous) reviewers. Background and foreground may differ in contrast, darkness or both. One reason for face to be perceived feminine (and ED-positive) is to possess a texture (a pattern of foreground noise) that makes features more female-like (while keeping darkness and root-mean-scale of contrast constant in background and foreground). The other reason that some face would be perceived more feminine (and ED-positive) is that the noise in the foreground area was lighter than the noise in the background (or there were differences in RMS contrast between foreground and background). Therefore, I find it unnecessary to run such analyses. Authors should make the section shorter or transfer it into supplementary materials. It seems too long compared with the rest of the paper. It only tells: ‘The noise of the background/foreground of more feminine/ED-positive stimuli may be per se different from the noise in the background/foreground of less feminine/ED-negative stimuli'. It is, in my opinion, an inevitable property of the set of “noises” (and human perception). Authors may e.g. sum the results of these analyses into a table. If they decide to keep the section here, they should add (i) the definition of the “darkness” authors apply (e.g. a reverse scale to the CIELab Lightness?); (ii) the description of the variables (mean ± SD, range). Also, the name of the used statistical programme and a brief description of the analysis should be included in the manuscript (and the analysis code should be provided on-line). Last but not least, interested readers could run much more analyses, and I predict their courage would enable them to run all the analyses they wish (as soon as they will obtain the data), there is no need to tell them so explicitly.

3. Secondary objections

The minor issues are as follows (in order of appearance in the text):

Page 2 (abstract, last four lines): Feminisation is of importance for ED diagnosis; however, when the noise is applied, perceived femininity is unimportant for ED diagnosis in white faces.

On page 3 of the manuscript, authors state that “Subclinical eating disorders symptoms that do not fully cross the clinical threshold for a disorder can also be problematic”. They should cite a relevant source for this claim. I miss a definition of subclinical eating disorders symptoms. As an evolutionary psychologist, I would expect them to exist and affect the perceived traits.

The college students should present future clinicians? Or, is there, due to difficulties with ED-diagnosing, a need for attentive students who recognise there is “something wrong” with a schoolmate?

The introduction, as a whole, seems to propose a “benchmark” study, which utilises the method of reverse correlation in an ecologically relevant context. However, the authors do not discuss (and justify) the psychological context in enough detail (see above).

Experiment 1

What does (on page 7) so-called “face space” stands for? I am aware of face space based upon facial landmarks and geometric morphometric (see, e.g.[9,10]). However, I would highly appreciate if the authors could bring a more detailed description of characteristics of FaceGen face space (e.g. a reference to previous studies that used FaceGen database). Also, the term “average faces” may describe a face that was attained by blending (or morphing) together with a set of faces [11,12]. Does “averageness” in FaceGen follow a similar method to create “average face”? In my opinion, there is no objection against using FaceGen stimuli; however, I cannot decide, whether the FaceGen faces “reflect systematic real-world category differences in face appearance” and whether “these capture sufficiently meaningful and representative appearance differences”. Are those claims based on some geometric morphometrics or other shape analyses? Otherwise, the authors should mention that using FaceGen faces may present a limitation (with regard to actual face diversity in the current US population).

Page 9-11: There is a full description of the analysis, and I think that authors use all the tests properly. Nonetheless, I miss the name of statistical software you use (e.g. “We did the test W using the function xyz in package zyx for R software (CITATION-OF-THE-PACKAGE)”).

Page 10 (and 15): To my knowledge, it is possible to obtain more detailed information about AMT workers who completed the tasks. Authors should describe their country of origin, sex and average age.

The data has been UNAVAILABLE from the link https://osf.io/kbmz7/ (even though I possess an OSF account, and I requested the permission). Therefore, I could not observe the data, calculate the missing variable descriptions (e.g. mean +/- SD and range of darkness and RMS contrast), neither run the binomial, chi-quadrat tests and ANOVAs on my own. This needs to be fixed before the article release.

Experiment 2 (page 15-17)

Authors should define the term “feminised” face in more detail. It may point either to (a) a result of artificial manipulation of faces based on scale connecting male and female facial configurations [9]; (b) a concept of perceived sex typicality, that may also be affected by non-shape cues [13]. Based upon information available, I predict that the authors followed the criteria and scale of femininity defined implicitly within the FaceGen database. However, I miss a definition of this scale: As mentioned previously, a “face space” is a rather diverse conceptual framework, which can be based upon 2D or 3D facial shape, skin contrast, colouration, and the allometric component of sexual dimorphism (and combination of those, like in FaceGen, most obviously). It is important to define the process of “feminisation” in more detail to allow other researchers to reply and extend the research based on other face databases. The statement that “varying faces according to FaceGen’s sense of femininity does correspond to meaningful appearance variation across male and female faces both in terms of 3D shape and 2D surface properties” is not sufficient. Authors should provide either (i) reference to the analysis of the 3D and 2D FaceGen faces’ properties which proves this; (ii) report of thier own analysis to support this claim.

Sometimes, authors do not follow formal rules for reporting the p-values: In the section “Image statistics analysis” (page 12), p = 0.035 is in neither case a “trend”. According to p = 0.035, you are allowed to “reject H0”.

In section “Experiment 2” (page 16-17) I miss a (two-tailed binomial) test to compare the proportions of trials in which feminised faces were chosen as looking more likely to have an ED diagnosis.

In the Discussion (page 17-18) authors should discuss in more detail the fact that artificially feminised faces of both White and Black race were perceived more likely as having an ED diagnosis. When noise applied, only in black faces was there a significant association between perceived femininity and ED-diagnosis. Perceived femininity (based upon noise) and manipulated femininity tell a different story (the latter being not affected by the race of the face).

The language and wordings are easily-accessible.

Finally, I would like to thank you for that the authors brought me an option to read the current version of the report on the study. I hope that the authors have been able to understand all my remarks and questions and that they are able to cope with them to present their research report to the scientific community.

References:

1. Stewart DE, Robinson GE, Goldbloom DS, Wright C. Infertility and eating disorders. Am J Obstet Gynecol. 1990;163: 1196–1199. doi:10.1016/0002-9378(90)90688-4

2. Smink FRE, Van Hoeken D, Hoek HW. Epidemiology of eating disorders: Incidence, prevalence and mortality rates. Curr Psychiatry Rep. 2012;14: 406–414. doi:10.1007/s11920-012-0282-y

3. Rhodes G, Zebrowitz LA, Clark A, Kalick SM, Hightower A, McKay R. Do facial averageness and symmetry signal health? Evol Hum Behav. 2001;22: 31–46. doi:10.1016/S1090-5138(00)00060-X

4. Nedelec JL, Beaver KM. Physical attractiveness as a phenotypic marker of health: An assessment using a nationally representative sample of American adults. Evol Hum Behav. 2014;35: 456–463. doi:10.1016/j.evolhumbehav.2014.06.004

5. Little AC, Jones BC, DeBruine LM. Facial attractiveness: evolutionary based research. Philos Trans R Soc B Biol Sci. 2011;366: 1638–1659. doi:10.1098/rstb.2010.0404

6. Henderson AJ, Holzleitner IJ, Talamas SN, Perrett DI. Perception of health from facial cues. Philos Trans R Soc B Biol Sci. 2016;371. doi:10.1098/rstb.2015.0380

7. Matts PJ, Fink B, Grammer K, Burquest M. Color homogeneity and visual perception of age, health, and attractiveness of female facial skin. J Am Acad Dermatol. 2007;57: 977–984. doi:10.1016/j.jaad.2007.07.040

8. Fink B, Grammer K, Matts PJ. Visible skin color distribution plays a role in the perception of age, attractiveness, and health in female faces. Evol Hum Behav. 2006;27: 433–442. doi:10.1016/j.evolhumbehav.2006.08.007

9. Mitteroecker P, Windhager S, Møller GB, Schaefer K. The morphometrics of “masculinity” in human faces. PLoS One. 2015;10: e0118374. doi:10.1371/journal.pone.0118374

10. Kleisner K, Pokorný Š, Saribay SA. Toward a new approach to cross-cultural distinctiveness and typicality of human faces: The cross-group typicality/ distinctiveness metric. Front Psychol. 2019;10: 1–13. doi:10.3389/fpsyg.2019.00124

11. Langlois JH, Roggman LA. Attractive Faces Are Only Average. Psychol Sci. 1990;1: 115–121. doi:https://doi.org/10.1111/j.1467-9280.1990.tb00079.x

12. Perrett DI, May KA, Yoshikawa S. Facial shape and judgements of female attractiveness. Nature. 1994;368: 239–242. doi:10.1038/368239a0

13. Komori M, Kawamura S, Ishihara S. Multiple mechanisms in the perception of face gender: Effect of sex-irrelevant features. J Exp Psychol Hum Percept Perform. 2011;37: 626–633. doi:10.1037/a0020369

6. PLOS authors have the option to publish the peer review history of their article (what does this mean?). If published, this will include your full peer review and any attached files.

Reviewer #1: **Yes: **Vojtěch Fiala

---

## [Author Response · Author response to Decision Letter 0]

8 Jan 2021

Please see the "Response to Reviews" document that we have uploaded along with our other manuscript files.

---

## [Decision Letter · Decision Letter 1]

27 Apr 2021

PONE-D-20-16005R1

Facial Femininity and Perceptions of Eating Disorders: A Reverse-Correlation Study

PLOS ONE

Dear Dr. Balas,

Thank you for submitting your manuscript to PLOS ONE. After careful consideration, we feel that it has merit but does not fully meet PLOS ONE’s publication criteria as it currently stands. Therefore, we invite you to submit a revised version of the manuscript that addresses the points raised during the review process.

The manuscript has been further evaluated following your initial revisions, and the reviewers' comments are available below.

The reviewers note that you have largely addressed their previous concerns with sufficient explanation and revision. However, there are still a few minor points which they request that you address, including further explanation in the Discussion regarding the relationship between the current and previous results. They also encourage you to more fully explain terms and abbreviations prior to usage, as well as include all references and maintain a consistent citation style.

Could you please revise the manuscript to carefully address the concerns raised?

We look forward to receiving your revised manuscript.

Kind regards,

Avanti Dey, PhD

Staff Editor

PLOS ONE

Journal Requirements:

Reviewers' comments:

Reviewer's Responses to Questions

**Comments to the Author**

1. If the authors have adequately addressed your comments raised in a previous round of review and you feel that this manuscript is now acceptable for publication, you may indicate that here to bypass the “Comments to the Author” section, enter your conflict of interest statement in the “Confidential to Editor” section, and submit your "Accept" recommendation.

Reviewer #1: (No Response)

Reviewer #2: All comments have been addressed

Reviewer #3: All comments have been addressed

2. Is the manuscript technically sound, and do the data support the conclusions?

Reviewer #1: Yes

Reviewer #2: Yes

Reviewer #3: Yes

3. Has the statistical analysis been performed appropriately and rigorously? 

Reviewer #1: Yes

Reviewer #2: Yes

Reviewer #3: Yes

4. Have the authors made all data underlying the findings in their manuscript fully available?

Reviewer #1: Yes

Reviewer #2: Yes

Reviewer #3: Yes

5. Is the manuscript presented in an intelligible fashion and written in standard English?

Reviewer #1: Yes

Reviewer #2: Yes

Reviewer #3: Yes

6. Review Comments to the Author

Reviewer #1: 1.Summary

Commentary on the revised version of the research report "Facial Femininity and Perception of Eating Disorders: A Reverse-Correlation Study". In the current version, the author did not substantially change the layout of the manuscript; therefore, the research report's description would be very much the same. Briefly, the authors used reverse correlation and FaceGen stimuli database to study whether facial femininity affects the probability that the face would be classified as belonging to a person who suffers from Eating Disorders (ED+ when yes, ED- when not). Authors also studied whether such ascribed characteristic is affected by the race (White or Black) of the face (stimuli). They found that when noise is applied, only faces of the Black race perceived as ED+ are also perceived as more feminine (by an independent group of on-line raters). Therefore, the authors demonstrated that race affects the way raters utilize facial cues when assessing eating disorders. In the second study, however, when they used faces manipulated in facial femininity, both the Black and White race stimuli that were feminized, were more likely classified ad ED+.

The authors highlight the importance of their study as a "cornerstone" of a promising research field. The field would study whether facial appearance may affect the probability ("likelihood") of a person being perceived (and eventually mistreated) as likely or unlikely "victim" of an eating disorder, based solely on their appearance. They show that being of certain origin may affect the probability of being perceived as suffering from a certain illness (ED, in this very case). Authors also took the first step in identifying the traits that serve in such "folk classification" (sex-typicality).

2. Review of revisions

The authors took all my suggestions and objections into consideration and either extended/shortened the manuscript to address them, or replied why I might have been wrong. I appreciate their patience. Therefore, I recommend the paper for acceptance because, after the revisions, I can more clearly see its potential impact. The authors also state their willingness to proceed on this topic.

They added or extended the following sections within the manuscript:

They re-wrote the abstract to emphasize that they studied "perceptual biases regarding facial appearance" and if they affect the diagnosis based on the race of the stimuli face. They also extended the introduction to stress that they studied (racially-specific) bias itself and did not identify the alleged "kernel of truth" in such a classification.

Throughout the description of methods, the authors also extended the section on the FaceGen database. I appreciate that. The extended description substantially improved my image on the utility of the database.

The authors also edited the section on the image analysis. While they kept the section on the effect of image background/foreground colouration, they shortened the section on the effect of root mean square contrast.

In experiment 2, the authors added another short description of the stimuli processing within FaceGen. Authors also added the ethics statement and changed the term "Eating disorder" to ED (and of course, they explained the abbreviation).

Authors solved the technical issue of data-inaccessibility (regardless of whether it was a failure of mine or the OSF itself). They also cited the statistic software they use so that I could check the data and analyses. However, in my opinion, they only provide the JASP files for the Image statistical analysis. For the rest of the analyses, I could find only the raw MTurk data and classification images.

Coming to the Response to Reviews section, authors replied in detail to my notions and suggestions, bringing further justification to their changes (or their decisions not to change the text according to the requests).

I accept their reply on my primary objections: I would like to thank them for explaining the study's actual goal and demonstrating that I was wrong in my suggestions. They explain what it is not a good idea to link the paper to evolutionary psychology. The revised version is much clearer regarding studying perceptual stereotypes based on facial traits and race (see also above).

I also accept that it would be extraordinarily complicated and ethically questionable to use photos of actual patients who suffer from eating disorders. In light of the manuscript's update, my suggestions on the use of different stimuli and study design are no more relevant. Also, the authors are right that reverse correlation (RC) is not a rare technique. My notion regarded just its usage within the field of face perception. Now, it is much clearer that both the goals and the whole "paradigm" of the current research report are much further from evolutionary psychology that I've been thinking. RC, therefore, seems a useful approach within the study of perceptual biases.

They also claim that my suggestion to identify which features of the classification image affect the classification is partly resolved by the Image statistical analyses. Otherwise, they argue that capturing the aspects of texture that led to stimuli classification is above the article's scope of the article and exceeds the capacity and abilities of image analysis.

3. Suggestions:

Therefore, I have only several minor objections, which could, in my opinion, be solved easily:

Page 2: "...their femininity may affect observers' beliefs about their eating behaviour" I would add: "only under specific circumstances (manipulation of stimuli)". -> The abstract has been modified upon my request. My request, however, did not address the difference in results between the first and second experiment—my apologies.

Page 5/References: Authors should add the reference to the following citation: Birhane & Guest, 2020 (currently missing in References).

Page 8: "That is, given this particular sample of faces described in terms of their 3D shape and 2D surface properties, what is a compact dimensional model of the data that captures the correlational structure of the data? The resulting model makes it possible to assign spatial coordinates to each face such that groups of faces can be characterized in terms of their distribution within this space." The second sentence, in my opinion, do not answer the first question. Consider writing something like: "That is, given this particular sample of faces described in terms of their 3D shape and 2D surface properties, the compact dimensional model of the data based on the faces makes it possible to assign spatial coordinates to each face such that faces can be characterized in terms of their distribution within this space." Of course, if I miss the important message of the sentences, keep the original version.

Page 12: "...covaried with the appearance characteristics associated with disordered eating by our first group or observers." Did you mean "first group of observers"?

Page 12/References: Authors should add the reference to the following citation: JASP, 2018 (currently missing in References).

Page 20/References: Authors should add the reference to the following citations: Fink et al., 2006; Matts et al., 2007; Mitteroecker et al., 2015 (currently missing in References).

Throughout the text, authors should unify the citation of the American Psychological Association data (either 2015 or 2013).

Reviewer #2: The authors did an excellent job in addressing reviewer comments. I only have a few minor additional comments.

Abstract

Please briefly define what the reverse correlation technique is in the abstract

Instead of saying “above-change levels” please include the relevant statistics

Introduction

It is good that you explained the reverse correlation technique, but I would do it before you introduce the term. Perhaps refrain from introducing until it is explained.

I found the explanation of the reverse correlation technique to be confusing

Please elaborate why peer-identification of EDs is important and biases may be problematic – e.g., that peers could refer individuals to treatment,

Methods:

I would minimize the use of acronyms – e.g., say MTurk instead of AMT workers, reverse correlation instead of RC

Discussion:

Please discuss in more detail the prevalence in men vs. women with EDs (i.e., what percentage of EDs is in men vs. women? Does differ by the ED?)

Reviewer #3: Thank you for the opportunity to review this manuscript. This is an interesting topic that can be considered by readers. The Introduction section was very well prepared and comprehensively provides what is already known on this subject. The results are very interesting. However, after reviewing the manuscript, I have some minor comments:

1. I would recommend expanding the Discussion section as to how some of the results of the study could be implemented rather than a repeat of what was already discussed.

2. References are missing in several places in the discussion.

3. How these results compare to previous results is not discussed.

7. PLOS authors have the option to publish the peer review history of their article (what does this mean?). If published, this will include your full peer review and any attached files.

Reviewer #1: **Yes: **Vojtěch Fiala

Reviewer #2: No

Reviewer #3: No

---

## [Author Response · Author response to Decision Letter 1]

4 May 2021

These responses are also included in the "Response to Reviews" document uploaded along with our revised manuscript files. 

Response to Reviews

We would like to thank all three of our reviewers for their thoughtful comments on the latest draft of our manuscript. We were pleased to see that they were very positive about this version of the text and we hope that these revisions have adequately addressed the issues they raised here. Below we have reproduced their narrative comments about the current draft, with our responses in boldface text.

Journal Requirements:

We have reviewed the reference list and incorporated the new citations suggested by our reviewers (noted below in response to their suggestions to do so). Otherwise, we have also corrected reference formatting in a few instances to be consistent with PLoS’ instructions to authors.

Reviewer's Responses to Questions

We have omitted the Reviewers’ responses to these questions in this document because there were no additional comments accompanying their answers. Below, we reproduce their comments to the authors in full, with point-by-point responses to each suggestion that indicate how we have edited the text to incorporate their feedback.

6. Review Comments to the Author

Reviewer #1: 1.Summary

Commentary on the revised version of the research report "Facial Femininity and Perception of Eating Disorders: A Reverse-Correlation Study". In the current version, the author did not substantially change the layout of the manuscript; therefore, the research report's description would be very much the same. Briefly, the authors used reverse correlation and FaceGen stimuli database to study whether facial femininity affects the probability that the face would be classified as belonging to a person who suffers from Eating Disorders (ED+ when yes, ED- when not). Authors also studied whether such ascribed characteristic is affected by the race (White or Black) of the face (stimuli). They found that when noise is applied, only faces of the Black race perceived as ED+ are also perceived as more feminine (by an independent group of on-line raters). Therefore, the authors demonstrated that race affects the way raters utilize facial cues when assessing eating disorders. In the second study, however, when they used faces manipulated in facial femininity, both the Black and White race stimuli that were feminized, were more likely classified ad ED+.

The authors highlight the importance of their study as a "cornerstone" of a promising research field. The field would study whether facial appearance may affect the probability ("likelihood") of a person being perceived (and eventually mistreated) as likely or unlikely "victim" of an eating disorder, based solely on their appearance. They show that being of certain origin may affect the probability of being perceived as suffering from a certain illness (ED, in this very case). Authors also took the first step in identifying the traits that serve in such "folk classification" (sex-typicality).

2. Review of revisions

The authors took all my suggestions and objections into consideration and either extended/shortened the manuscript to address them, or replied why I might have been wrong. I appreciate their patience. Therefore, I recommend the paper for acceptance because, after the revisions, I can more clearly see its potential impact. The authors also state their willingness to proceed on this topic.

They added or extended the following sections within the manuscript:

They re-wrote the abstract to emphasize that they studied "perceptual biases regarding facial appearance" and if they affect the diagnosis based on the race of the stimuli face. They also extended the introduction to stress that they studied (racially-specific) bias itself and did not identify the alleged "kernel of truth" in such a classification.

Throughout the description of methods, the authors also extended the section on the FaceGen database. I appreciate that. The extended description substantially improved my image on the utility of the database.

Thank you - we’re glad to hear that this provided useful context. 

The authors also edited the section on the image analysis. While they kept the section on the effect of image background/foreground colouration, they shortened the section on the effect of root mean square contrast.

In experiment 2, the authors added another short description of the stimuli processing within FaceGen. Authors also added the ethics statement and changed the term "Eating disorder" to ED (and of course, they explained the abbreviation).

Authors solved the technical issue of data-inaccessibility (regardless of whether it was a failure of mine or the OSF itself). They also cited the statistic software they use so that I could check the data and analyses. However, in my opinion, they only provide the JASP files for the Image statistical analysis. For the rest of the analyses, I could find only the raw MTurk data and classification images.

We apologize - these files should be available in our OSF repository at present. 

Coming to the Response to Reviews section, authors replied in detail to my notions and suggestions, bringing further justification to their changes (or their decisions not to change the text according to the requests).

I accept their reply on my primary objections: I would like to thank them for explaining the study's actual goal and demonstrating that I was wrong in my suggestions. They explain what it is not a good idea to link the paper to evolutionary psychology. The revised version is much clearer regarding studying perceptual stereotypes based on facial traits and race (see also above).

I also accept that it would be extraordinarily complicated and ethically questionable to use photos of actual patients who suffer from eating disorders. In light of the manuscript's update, my suggestions on the use of different stimuli and study design are no more relevant. Also, the authors are right that reverse correlation (RC) is not a rare technique. My notion regarded just its usage within the field of face perception. Now, it is much clearer that both the goals and the whole "paradigm" of the current research report are much further from evolutionary psychology that I've been thinking. RC, therefore, seems a useful approach within the study of perceptual biases.

We appreciate the reviewer’s thoughtful commentary about these issues and are glad to hear that our revisions helped clarify the key goals of our study.

They also claim that my suggestion to identify which features of the classification image affect the classification is partly resolved by the Image statistical analyses. Otherwise, they argue that capturing the aspects of texture that led to stimuli classification is above the article's scope of the article and exceeds the capacity and abilities of image analysis.

3. Suggestions:

Therefore, I have only several minor objections, which could, in my opinion, be solved easily:

Thank you for these additional comments - below we describe how we have edited the text to address the remaining suggestions.

Page 2: "...their femininity may affect observers' beliefs about their eating behaviour" I would add: "only under specific circumstances (manipulation of stimuli)". -> The abstract has been modified upon my request. My request, however, did not address the difference in results between the first and second experiment—my apologies.

We have revised this as suggested by the reviewer. 

Page 5/References: Authors should add the reference to the following citation: Birhane & Guest, 2020 (currently missing in References).

Thank you. We have included this in the revised references.

Page 8: "That is, given this particular sample of faces described in terms of their 3D shape and 2D surface properties, what is a compact dimensional model of the data that captures the correlational structure of the data? The resulting model makes it possible to assign spatial coordinates to each face such that groups of faces can be characterized in terms of their distribution within this space." The second sentence, in my opinion, do not answer the first question. Consider writing something like: "That is, given this particular sample of faces described in terms of their 3D shape and 2D surface properties, the compact dimensional model of the data based on the faces makes it possible to assign spatial coordinates to each face such that faces can be characterized in terms of their distribution within this space." Of course, if I miss the important message of the sentences, keep the original version.

Thank you. We have adopted this suggestion in the revised text. 

Page 12: "...covaried with the appearance characteristics associated with disordered eating by our first group or observers." Did you mean "first group of observers"?

We did. Thank you! This has been corrected in the revised text.

Page 12/References: Authors should add the reference to the following citation: JASP, 2018 (currently missing in References).

We have added this to the references. 

Page 20/References: Authors should add the reference to the following citations: Fink et al., 2006; Matts et al., 2007; Mitteroecker et al., 2015 (currently missing in References).

We have added these to the references as well.

Throughout the text, authors should unify the citation of the American Psychological Association data (either 2015 or 2013).

We have corrected these inconsistencies. 

Reviewer #2: The authors did an excellent job in addressing reviewer comments. I only have a few minor additional comments.

Abstract

Please briefly define what the reverse correlation technique is in the abstract

We have included a short description of the technique in the revised abstract. 

Instead of saying “above-change levels” please include the relevant statistics

Our understanding is that specific statistical results are usually reserved for the main text of the paper, while the abstract is meant to serve as a more concise description of key outcomes. As such, we have edited the abstract to more clearly communicate this result, but only report the statistical tests we used and their outcomes in the relevant results section.

Introduction

It is good that you explained the reverse correlation technique, but I would do it before you introduce the term. Perhaps refrain from introducing until it is explained.

We have edited the text to minimize references to the technique before we explain its use. 

I found the explanation of the reverse correlation technique to be confusing

We apologize for the lack of clarity. We have attempted to re-work this part of the manuscript to make it more clear how reverse correlation paradigms work and why it is useful in this case.

Please elaborate why peer-identification of EDs is important and biases may be problematic – e.g., that peers could refer individuals to treatment,

We have attempted to expand upon this point in the revised draft.

Methods:

I would minimize the use of acronyms – e.g., say MTurk instead of AMT workers, reverse correlation instead of RC

We have adopted this suggestion throughout the text.

Discussion:

Please discuss in more detail the prevalence in men vs. women with EDs (i.e., what percentage of EDs is in men vs. women? Does differ by the ED?)

This does vary by eating disorder - we have briefly included some of these statistics in the revised text.

Reviewer #3: Thank you for the opportunity to review this manuscript. This is an interesting topic that can be considered by readers. The Introduction section was very well prepared and comprehensively provides what is already known on this subject. The results are very interesting. However, after reviewing the manuscript, I have some minor comments:

1. I would recommend expanding the Discussion section as to how some of the results of the study could be implemented rather than a repeat of what was already discussed.

Thank you for this comment, we have attempted to briefly offer some speculations along these lines in the discussion.

2. References are missing in several places in the discussion.

We have included these in the revised text.

3. How these results compare to previous results is not discussed.

We have attempted to clarify how these results relate to previous findings, though to some extent this is a relatively new avenue of inquiry regarding how facial appearance and the perception of eating disorder diagnoses may be correlated.

---

## [Decision Letter · Decision Letter 2]

8 Jul 2021

PONE-D-20-16005R2

Facial Femininity and Perceptions of Eating Disorders: A Reverse-Correlation Study

PLOS ONE

Dear Dr. Benjamin Balas,

Thank you for submitting your manuscript to PLOS ONE. After careful consideration, we feel that it has merit but does not fully meet PLOS ONE’s publication criteria as it currently stands. Therefore, we invite you to submit a revised version of the manuscript that addresses the points raised during the review process.

We look forward to receiving your revised manuscript.

Kind regards,

Kamila Czepczor-Bernat

Academic Editor

PLOS ONE

Reviewers' comments:

Reviewer's Responses to Questions

**Comments to the Author**

1. If the authors have adequately addressed your comments raised in a previous round of review and you feel that this manuscript is now acceptable for publication, you may indicate that here to bypass the “Comments to the Author” section, enter your conflict of interest statement in the “Confidential to Editor” section, and submit your "Accept" recommendation.

Reviewer #1: All comments have been addressed

Reviewer #4: All comments have been addressed

Reviewer #5: (No Response)

2. Is the manuscript technically sound, and do the data support the conclusions?

Reviewer #1: Yes

Reviewer #4: Partly

Reviewer #5: Yes

3. Has the statistical analysis been performed appropriately and rigorously? 

Reviewer #1: Yes

Reviewer #4: Yes

Reviewer #5: Yes

4. Have the authors made all data underlying the findings in their manuscript fully available?

Reviewer #1: Yes

Reviewer #4: Yes

Reviewer #5: Yes

5. Is the manuscript presented in an intelligible fashion and written in standard English?

Reviewer #1: Yes

Reviewer #4: Yes

Reviewer #5: Yes

6. Review Comments to the Author

Reviewer #1: The research report studied whether the diagnosis (perception) of eating disorder based solely on facial appearance is associated with facial femininity and whether the association is affected by race of the stimuli (Black or White). It used the reverse correlation technique.

In the previous (first) round of revisions, I already appreciated that the authors

(i) had improved and clarified any potentially misleading sections (e.g., explained that they did not study whether ED-diagnosing is anyhow accurate, described that they did not use a perspective of evolutionary psychology, and made clear that for serious reasons, they cannot use real ED-diagnosed individuals);

(ii) had extended section on the description of the reverse correlation technique;

(iii) had extended description on the FaceGen database, too.

After the second round of revisions, I have to confirm that none of those improvements was anyhow affected (worsened) by new modifications. Moreover, I appreciate that the section on the reverse correlation technique, which was good already, has been improved. I will remember that the paper contains such a section, and I would eventually refer to that when asked about the RC technique.

Furthermore, the authors added all missing references into the list and took my suggestions concerning several sentences, which were, in my opinion, misleading or hard to follow. The authors also uploaded potentially missing data to the online supplementary material. It allows following a "flow" of statistical analyses. Nonetheless, I have to declare that I am not an advanced user of the programs and tools that the authors used - JASP, Matlab, and MS Office Access. I am more familiar with R software.

Given that the paper focuses on two topics of increasing importance (eating disorders and racial stereotyping), concisely but sufficiently describes methods, reports and discuss the results correctly, and has been revised with regard to previously identified misleading or incorrect sections, I recommend it to final acceptance with no further revisions required. Apparent discrepancies in used font and poorly placed paragraphs within references would be, in my opinion, resolved by the final typeset in the journal.

Reviewer #4: Basically, the current research seems to be academically meaningful. However, it seems that there are parts that need correction and completion of the research.

First, what is the operational definition of perceptual bias in this study? I wonder how the authors can prove that what they observe in this study corresponds to perceptual bias.

Second, it is hard to understand what it means in reality for ordinary people to look at other people's faces and guess that they have eating disorders. In terms of diagnosis, clinically diagnosing eating disorders has no significant relationship with facial features. In particular, there are many forms of eating disorders, such as anorexia and bulimia, and it seems necessary to carefully explain how this potentially affected the research results.

Third, when participants are asked which of the two is more likely to have eating disorders by looking at the experimental stimulation faces, it is difficult to understand what criteria they made. How can we check if the participants understood the meaning of the task properly? Even if independent observers agreed with the results of in-lab participants, this does not address the question of what criteria they used in the evaluation process. Basically, it is a different task for the ordinary people to perform the task of choosing the more feminine of the two facial stimuli and the task of choosing the one with the potential for eating disorders.

Fourth, it seems difficult to interpret this study as providing evidence that “facial appearance may affect decisionmaking related to the estimated likelihood of health-related behaviors in others, in this case, disordered eating.” If another group of researchers constructed a research design to confirm the evidence that facial appearance can affect decisionmaking related to the estimated likelihood of eating disorders, I wonder if they would construct the same design as this study. The findings may not be a study of decision-making related to the estimated likelihood of eating disorders but simply reflecting stereotypes shared by ordinary people.

In conclusion, although this is an interesting study, it seems necessary to refine the purpose and meaning of the study by considering the limitations of the study.

Reviewer #5: Referee report “Facial Femininity and Perceptions of Eating Disorders: A Reverse-Correlation Study” PONE-D-20-16005R2

Note: I use singular author throughout to refer to author/author(s)

Summary

This paper reports on two experiments (with three separate groups of participants involved) that elicited people to make choices regarding subjective evaluations of the appearance of faces, following a prompt. One experiment was a reverse correlation task (choosing between noise-adjusted face pairs according to a prompt), the other asked participants to characterize experimenter-manipulated face images (choosing between original and adjusted face according to a prompt). The faces and prompts varied on three dimensions: eating disorders (ED), gender, and race (white or black).

The author variously frames the main finding (or key hypothesis to be evaluated) as:

• “We hypothesized that observers would believe that faces that looked more feminine would also be more likely to have an ED diagnosis. Further, we hypothesized that this tendency might be affected by the race of the face under consideration due to additional observer biases affecting how femininity is perceived across racial categories.”

• “In particular, we examined whether or not perceived femininity was differentially linked to perceived likelihood of having an eating disorder diagnosis as a function of race.”

• “How does femininity co-vary with image features related to the perception of disordered eating, and does this relationship vary by race?”

The results are then described as:

• “We found that for Black faces, perceived femininity was associated with the perceived likelihood of having an eating disorder diagnosis. In our second experiment, directly manipulating femininity to influence eating disorder categorization did not lead to differences in the outcome for Black and White faces, though we did observe a trend favoring a larger impact of feminization on the rate at which feminized Black faces were categorized as more likely to have an eating disorder diagnosis.”

• “For Black faces, on the other hand, since their race violates the stereotype of who is more likely to have an eating disorder, assessors may tend to focus more on the relevant characteristic they have that goes with the stereotype - being female. As being female becomes the characteristic of focus for Black faces, an increase in facial femininity leads to a stronger association with eating disorder.”

The article is motivated by the observation that a distinct population, college students, on average neither recognizes ED symptoms, nor the need for ED treatment, nor is able to “consistently identify what constitutes disordered eating”. Moreover, college students (?) likely have racial stereotypes of EDs.

Overall Evaluation

The paper is well-written, the methods are appropriate, the analysis is competent. Authors offer a clear and concise presentation. The findings are somewhat mixed: at the end this reader was not sure whether to come away thinking, “race definitely matters in mediating people’s association with ED and femininity” or to come away thinking “probably race does not matter much in mediating people’s association with ED and femininity.”

A significant missing element in the protocol/data is an identifier of race for the experiment respondents: while Black v. white face might condition the correlation between femininity and ED, presumably the race-identification of the participants (as Black or white, or other ascribed or self-ascribed ethnic identities) would matter significantly. Would a population of Black persons likely have a correlation between femininity and ED similar to that of a population of white persons, for images of Black and white faces? This strikes me as unlikely, and yet the identities of the participants are not mentioned (only their gender is noted but not used as a covariate in the analysis). Perhaps there were no Black students in the college sample? The Amazon MechTurk tasks in Experiment 1 and Experiment 2 were done by respondents (unclear how many), and their identity status appears not to be a covariate in the analysis.

A related area of ambiguity (for this reader) was who was expected to be the population about which the research could be generalized. In first paragraph, this appears to be college students, but then the second paragraph mentions clinicians having stereotypes. Then bottom of p. 3 becomes more general, mentioning “people.” On p. 6 and 7 the relevant population is called “naïve observers.” Then in methods, the Experiment 1 RC task was done by 28 college students from North Dakota (no mention of how identity backgrounds might matter except to mention that clinicians were excluded). The two Amazon MechTurk samples were presumably not college students. So it is unclear who constitutes the population of interest. Amazon MechTurk are hardly a random sample of a well-defined population, and author should perhaps consult the expanding literature on how to generalize from Amazon MechTurk samples.

Presentation of results (some questions and suggestions for improvement)

There are essentially three results in the paper.

1. “The [28 pairs of] classification images created by in-lab participants were thus reliably interpreted by independent observers as looking more likely to have an eating disorder across both White and Black faces, but race affected how perceived femininity covaried with the appearance characteristics associated with disordered eating by our first group of observers.”

a. Unclear here whether 560 participants each scored one pair? Or whether 40 Amazon MechTurk participants each scored the 28 pairs? The paper says, “We collected a total of 560 independent judgments for each question for both.” This makes it sound like 560 different participants? Is there no information about these participants?

b. White and Black classification images +ED were correlated with the evaluation of separate raters that +ED were somewhat more likely to have an eating disorder. Black +ED faces were more likely to be perceived as more feminine than –ED pair.

2. The differences in reverse correlation +ED were not due to differences in background or face luminance (and nice validation exercise).

a. There appears to be a mistake in the text: on p. 15 “(t (27) = -2.21, p = 0.035)” so normally this indicates rejecting the null hypothesis, but the text states “nor Black face had systematically darker backgrounds for ED+ face images compared to ED- face images”

b. One might wonder though whether “mottling” of face is/was relevant for ED choice?

3. 400 Amazon Mechanical Turk workers each examined one image pair and rated it. The raters preponderately found that the femininized faces were more likely to have an eating order diagnosis. Black and white faces were only marginally different from each other.

Smaller questions

Fig. 2 – The average classification images – is this average of the averages- just to be clear? Or is it one of the 28 individual CI?

How many of Amazon MechTurk who did task may have been repeat workers? That is, could they not have multiple accounts and so the same person did many responses?

Comment

More modesty about what RC does? “the experimenter can obtain an image that reflects what properties of the noise were correlated with perceived membership in that category.” Could be “the experimenter can obtain an image that might possibly reflect some patterns of the noise correlated with perceived membership in that category.” (the possibly is because some of the correlations might be spurious as there are very large number of patterns (i.e., quite multidimensional) and simple patterns might be correlated when the “true” pattern that generates correlation is more complex.

7. PLOS authors have the option to publish the peer review history of their article (what does this mean?). If published, this will include your full peer review and any attached files.

Reviewer #1: **Yes: **Vojtěch Fiala

Reviewer #4: No

Reviewer #5: No

---

## [Author Response · Author response to Decision Letter 2]

10 Jul 2021

Response to reviews:

We would like to thank the editor and the reviewers for their further comments on our revised manuscript. We are pleased to see that two of our reviewers indicated that the manuscript was acceptable for publication in its current form and we have incorporated the additional comments made by all of the reviewers into this current draft. Below, we respond to each of these comments individually to highlight how we have edited the text to address these remaining issues. The original review comments are in quotes and our responses follow throughout.

Reviewer's Responses to Questions

"Reviewer #1: The research report studied whether the diagnosis (perception) of eating disorder based solely on facial appearance is associated with facial femininity and whether the association is affected by race of the stimuli (Black or White). It used the reverse correlation technique.

In the previous (first) round of revisions, I already appreciated that the authors

(i) had improved and clarified any potentially misleading sections (e.g., explained that they did not study whether ED-diagnosing is anyhow accurate, described that they did not use a perspective of evolutionary psychology, and made clear that for serious reasons, they cannot use real ED-diagnosed individuals);

(ii) had extended section on the description of the reverse correlation technique;

(iii) had extended description on the FaceGen database, too.

After the second round of revisions, I have to confirm that none of those improvements was anyhow affected (worsened) by new modifications. Moreover, I appreciate that the section on the reverse correlation technique, which was good already, has been improved. I will remember that the paper contains such a section, and I would eventually refer to that when asked about the RC technique.

Furthermore, the authors added all missing references into the list and took my suggestions concerning several sentences, which were, in my opinion, misleading or hard to follow. The authors also uploaded potentially missing data to the online supplementary material. It allows following a "flow" of statistical analyses. Nonetheless, I have to declare that I am not an advanced user of the programs and tools that the authors used - JASP, Matlab, and MS Office Access. I am more familiar with R software.

Given that the paper focuses on two topics of increasing importance (eating disorders and racial stereotyping), concisely but sufficiently describes methods, reports and discuss the results correctly, and has been revised with regard to previously identified misleading or incorrect sections, I recommend it to final acceptance with no further revisions required. Apparent discrepancies in used font and poorly placed paragraphs within references would be, in my opinion, resolved by the final typeset in the journal."

Thank you very much for these comments! We are very glad to see that our revisions addressed the comments this reviewer made in previous rounds of review.

Reviewer #4: Basically, the current research seems to be academically meaningful. However, it seems that there are parts that need correction and completion of the research.

"First, what is the operational definition of perceptual bias in this study? I wonder how the authors can prove that what they observe in this study corresponds to perceptual bias."

Our operational definition of perceptual bias in the current study is observers’ tendency to link specific aspects of facial appearance to category labels: In this study, we are specifically interested in the link between perceived femininity (one aspect of facial appearance that is determined in part by pigmentation and contrast relationships in the face) and the categorization of faces according to the likelihood of an eating disorder diagnosis. As to whether or not we can truly “prove” that our results are the result of a perceptual bias, we would argue that proving causal mechanisms for behavioral results is generally extremely difficult. What we offer here (and take more care in the revised text to clarify) is evidence that observers appear to have different expectations about the covariation between feminine appearance and eating disorder likelihood as a function of stimulus race. Whether the causal mechanisms leading to this outcome are uniquely perceptual or cognitive, our task reveals that these expectations affect performance in a perceptual task. In the revised text, we have attempted to be more careful about our conclusions in this regard which we hope adequately addresses the reviewer’s point.

"Second, it is hard to understand what it means in reality for ordinary people to look at other people's faces and guess that they have eating disorders. In terms of diagnosis, clinically diagnosing eating disorders has no significant relationship with facial features. In particular, there are many forms of eating disorders, such as anorexia and bulimia, and it seems necessary to carefully explain how this potentially affected the research results."

As we have described in the revised introduction, observers readily estimate a number of personality variables from face images despite many of these judgments having low validity. Across a large literature, observers have been demonstrated to make reliable judgments about trustworthiness, dominance, competence, and other aspects of mental health from face images, which we think indicates that this is indeed an aspect of face recognition that is relevant to real-world behavior and leads to quantifiable outcomes in a range of tasks that can be used to make inferences about the basis for those judgments. The reviewer is correct that there are not known links between facial appearance and eating disorder symptomology (which we are careful to point out in the text, contrasting our study with what has been dubbed “modern physiognomy” by experts in the ethics of face recognition), but this does not mean observers do not have expectations about correlations between appearance and behavior. We agree with the reviewer that the range of eating disorders could potentially be interesting to consider, but we chose to ask observers to make a judgment about disordered eating considered broadly to avoid the need to explain differences between, say, anorexia nervosa and binge eating disorder that might influence participants’ choices in our task. In the revised text we have edited the text to speak to both of these issues and make clear that it is not evidence of a valid link between facial appearance and disordered eating that we are pursuing, but rather observers’ potential beliefs about the same. 

"Third, when participants are asked which of the two is more likely to have eating disorders by looking at the experimental stimulation faces, it is difficult to understand what criteria they made. How can we check if the participants understood the meaning of the task properly? Even if independent observers agreed with the results of in-lab participants, this does not address the question of what criteria they used in the evaluation process. Basically, it is a different task for the ordinary people to perform the task of choosing the more feminine of the two facial stimuli and the task of choosing the one with the potential for eating disorders."

We respectfully disagree with the reviewer on this point. The reverse correlation technique specifically allows us to examine what aspects of facial appearance the observers were using to make this decision: The individual classification images show us what image templates correlate with the ED+ and ED- judgments per observer, and the independent ratings made by our additional participants allow us to examine how high-level variables like femininity also correlate with these judgments. Further, we suggest that the agreement between our two samples of participants regarding the likelihood of an eating disorder diagnosis in ED+ and ED- classification images is indeed a useful demonstration that both participant groups understood the task and understood it in similar ways. We also see no evidence in our data (or from post-testing debriefing) that participants did not understand the task properly: Observers’ classification images were categorized by independent observers at above-chance levels, suggesting that the image features used by both groups to assess the likelihood of an eating disorder were similar, and we also observed differences in specific image statistics across ED+ and ED- categories. Observers thus made reliable image-based judgments that led to measurable effects across our target categories. This 2AFC judgment is thus not any more suspect than other 2AFC judgments about facial appearance, especially when one considers the other examples of social categorization task we now describe in the revised introduction. While we agree with the reviewer that estimating the likelihood of an ED diagnosis is “a different task” than categorizing gender, it is also a different task that still leads to consistent behavior within and between individuals and did not lead our participants to express confusion with task requirements. 

"Fourth, it seems difficult to interpret this study as providing evidence that “facial appearance may affect decisionmaking related to the estimated likelihood of health-related behaviors in others, in this case, disordered eating.” If another group of researchers constructed a research design to confirm the evidence that facial appearance can affect decisionmaking related to the estimated likelihood of eating disorders, I wonder if they would construct the same design as this study. The findings may not be a study of decision-making related to the estimated likelihood of eating disorders but simply reflecting stereotypes shared by ordinary people."

Again, we respectfully disagree with the reviewer on this point. First, while we do not know if a hypothetical group of other researchers would come up with the same study design as we have used here, we also do not think this is a meaningful way to judge whether our work provides useful data or not. Different approaches to examine the same issue are valuable and we think this technique offers interesting insights that other designs would not. Indeed, as we describe in the introduction, reverse correlation tasks like this one have been used to establish similar links between facial appearance and various categorization judgments in many different cases. Our study is thus an application of an existing technique to a new problem domain, which we think does indeed address whether observers have expectations about how facial appearance may be related to health behaviors. We apologize, but we also do not understand distinction the reviewer makes between stereotypes shared by ordinary people (specifically with regard to what people who exhibit certain behaviors look like) and a link between facial appearance and decision-making. By the latter, we are simply referring to the outcomes we observed in our study: Observers made reliable decisions about behavior (eating disorders) based on what faces look like, which we think could reasonably be called a stereotype. We have assumed in preparing the revised text that the reviewer may have objected to our particular choice of words, and so we have edited the text to link our original wording to the language of stereotypes suggested by the reviewer. We hope this adequately addresses this point. 

In conclusion, although this is an interesting study, it seems necessary to refine the purpose and meaning of the study by considering the limitations of the study.

Reviewer #5: Referee report “Facial Femininity and Perceptions of Eating Disorders: A Reverse-Correlation Study” PONE-D-20-16005R2

Note: I use singular author throughout to refer to author/author(s)

Summary

This paper reports on two experiments (with three separate groups of participants involved) that elicited people to make choices regarding subjective evaluations of the appearance of faces, following a prompt. One experiment was a reverse correlation task (choosing between noise-adjusted face pairs according to a prompt), the other asked participants to characterize experimenter-manipulated face images (choosing between original and adjusted face according to a prompt). The faces and prompts varied on three dimensions: eating disorders (ED), gender, and race (white or black).

The author variously frames the main finding (or key hypothesis to be evaluated) as:

• “We hypothesized that observers would believe that faces that looked more feminine would also be more likely to have an ED diagnosis. Further, we hypothesized that this tendency might be affected by the race of the face under consideration due to additional observer biases affecting how femininity is perceived across racial categories.”

• “In particular, we examined whether or not perceived femininity was differentially linked to perceived likelihood of having an eating disorder diagnosis as a function of race.”

• “How does femininity co-vary with image features related to the perception of disordered eating, and does this relationship vary by race?”

The results are then described as:

• “We found that for Black faces, perceived femininity was associated with the perceived likelihood of having an eating disorder diagnosis. In our second experiment, directly manipulating femininity to influence eating disorder categorization did not lead to differences in the outcome for Black and White faces, though we did observe a trend favoring a larger impact of feminization on the rate at which feminized Black faces were categorized as more likely to have an eating disorder diagnosis.”

• “For Black faces, on the other hand, since their race violates the stereotype of who is more likely to have an eating disorder, assessors may tend to focus more on the relevant characteristic they have that goes with the stereotype - being female. As being female becomes the characteristic of focus for Black faces, an increase in facial femininity leads to a stronger association with eating disorder.”

The article is motivated by the observation that a distinct population, college students, on average neither recognizes ED symptoms, nor the need for ED treatment, nor is able to “consistently identify what constitutes disordered eating”. Moreover, college students (?) likely have racial stereotypes of EDs.

"Thank you for this summary – this indeed largely captures the design and results of our study."

"Overall Evaluation

The paper is well-written, the methods are appropriate, the analysis is competent. Authors offer a clear and concise presentation. The findings are somewhat mixed: at the end this reader was not sure whether to come away thinking, “race definitely matters in mediating people’s association with ED and femininity” or to come away thinking “probably race does not matter much in mediating people’s association with ED and femininity.”

"A significant missing element in the protocol/data is an identifier of race for the experiment respondents: while Black v. white face might condition the correlation between femininity and ED, presumably the race-identification of the participants (as Black or white, or other ascribed or self-ascribed ethnic identities) would matter significantly. Would a population of Black persons likely have a correlation between femininity and ED similar to that of a population of white persons, for images of Black and white faces? This strikes me as unlikely, and yet the identities of the participants are not mentioned (only their gender is noted but not used as a covariate in the analysis). Perhaps there were no Black students in the college sample? The Amazon MechTurk tasks in Experiment 1 and Experiment 2 were done by respondents (unclear how many), and their identity status appears not to be a covariate in the analysis."

We apologize for this omission – the sample of students who completed the classification image experiment was predominantly comprised of white students, which we include in the revised draft. We agree that including participant race as a covariate would be an excellent analysis, recruiting a sufficient sample of Black students is typically challenging at our institution given the demographics of the student population. With regard to our Mechanical Turk experiments, we chose not to collect race information from these participants. In the revised text we make this explicit, clarify the number of participants in each task, and discuss how observer race could affect these judgments. 

"A related area of ambiguity (for this reader) was who was expected to be the population about which the research could be generalized. In first paragraph, this appears to be college students, but then the second paragraph mentions clinicians having stereotypes. Then bottom of p. 3 becomes more general, mentioning “people.” On p. 6 and 7 the relevant population is called “naïve observers.” Then in methods, the Experiment 1 RC task was done by 28 college students from North Dakota (no mention of how identity backgrounds might matter except to mention that clinicians were excluded). The two Amazon MechTurk samples were presumably not college students. So it is unclear who constitutes the population of interest. Amazon MechTurk are hardly a random sample of a well-defined population, and author should perhaps consult the expanding literature on how to generalize from Amazon MechTurk samples."

We apologize for the confusion. First, our brief mention of clinicians was not intended to be a claim about the generality of our results, but rather a broader statement about why understanding the nature of these biases could be relevant to real-world contexts. Reviewer #4, for example, doubted whether or not estimating the likelihood of an eating disorder diagnosis from appearance was even a reasonable task to imagine, and we suggest that the clinical context is an important way to respond to this question. Second, Experiment 1 was indeed completed using a sample of college students, with no particular aspects of “identity background” constrained or measured. While we agree that the use of AMT in our subsequent tasks does represent a different (and likely more variable) population, this is increasingly common across a range of cognitive tasks and we think does not pose any major challenges to our main conclusions. We do agree that it is important to highlight that the judgments made in Experiment 1 were made by a specific participant group (college students in the US) and this may not be representative of the performance we might see in a different or broader group. The revised text includes a discussion of this point. We also discuss what conclusions we can draw from our AMT sample with regard to generality, trying to emphasize a conservative interpretation of our results. 

Presentation of results (some questions and suggestions for improvement)

There are essentially three results in the paper.

1. “The [28 pairs of] classification images created by in-lab participants were thus reliably interpreted by independent observers as looking more likely to have an eating disorder across both White and Black faces, but race affected how perceived femininity covaried with the appearance characteristics associated with disordered eating by our first group of observers.”

"a. Unclear here whether 560 participants each scored one pair? Or whether 40 Amazon MechTurk participants each scored the 28 pairs? The paper says, “We collected a total of 560 independent judgments for each question for both.” This makes it sound like 560 different participants? Is there no information about these participants?"

Yes, we asked for 560 independent judgments and did not collect demographic data. While this means we cannot include covariates in our analysis, our results simply collapse across whatever variability is in the data to describe the main results as simply as possible. 

". White and Black classification images +ED were correlated with the evaluation of separate raters that +ED were somewhat more likely to have an eating disorder. Black +ED faces were more likely to be perceived as more feminine than –ED pair.

2. The differences in reverse correlation +ED were not due to differences in background or face luminance (and nice validation exercise).

a. There appears to be a mistake in the text: on p. 15 “(t (27) = -2.21, p = 0.035)” so normally this indicates rejecting the null hypothesis, but the text states “nor Black face had systematically darker backgrounds for ED+ face images compared to ED- face images”

We would like to be cautious in terms of our interpretation of this difference given the multiple comparisons we included here. Applying a Bonferroni correction our data requires an alpha of 0.025 rather than 0.05, which means this result does not reach significance. We have clarified this in the revised text. 

"b. One might wonder though whether “mottling” of face is/was relevant for ED choice?"

This is an interesting issue, but difficult for us to address meaningfully using the current data. 

3. 400 Amazon Mechanical Turk workers each examined one image pair and rated it. The raters preponderately found that the femininized faces were more likely to have an eating order diagnosis. Black and white faces were only marginally different from each other.

Smaller questions

"Fig. 2 – The average classification images – is this average of the averages- just to be clear? Or is it one of the 28 individual CI?"

Yes, this is an average of averages. 

"How many of Amazon MechTurk who did task may have been repeat workers? That is, could they not have multiple accounts and so the same person did many responses?"

We do not know a clear way to determine whether or not someone created multiple accounts to carry out different judgments. We suggest this is another reason to be skeptical of our ability to collect meaningful demographic data about our AMT workers and is part of the reason we chose to collect and analyze this data in the aggregate. 

Comment

More modesty about what RC does? “the experimenter can obtain an image that reflects what properties of the noise were correlated with perceived membership in that category.” Could be “the experimenter can obtain an image that might possibly reflect some patterns of the noise correlated with perceived membership in that category.” (the possibly is because some of the correlations might be spurious as there are very large number of patterns (i.e., quite multidimensional) and simple patterns might be correlated when the “true” pattern that generates correlation is more complex.

We have added the word “possibly” to address this important point.

Once again we would like to thank our reviewers for their continued commentary on this 3rd revision of our manuscript. We hope that our revisions here have addressed their concerns and appreciate their constructive comments. We feel that the manuscript has been further strengthened by their thoughtful questions and observations.

---

## [Decision Letter · Decision Letter 3]

26 Jul 2021

Facial Femininity and Perceptions of Eating Disorders: A Reverse-Correlation Study

PONE-D-20-16005R3

Dear Dr. Benjamin Balas,

We’re pleased to inform you that your manuscript has been judged scientifically suitable for publication and will be formally accepted for publication once it meets all outstanding technical requirements.

Kind regards,

Kamila Czepczor-Bernat

Guest Editor

PLOS ONE

Additional Editor Comments (optional):

Reviewers' comments:

Reviewer's Responses to Questions

**Comments to the Author**

1. If the authors have adequately addressed your comments raised in a previous round of review and you feel that this manuscript is now acceptable for publication, you may indicate that here to bypass the “Comments to the Author” section, enter your conflict of interest statement in the “Confidential to Editor” section, and submit your "Accept" recommendation.

Reviewer #4: All comments have been addressed

Reviewer #5: All comments have been addressed

2. Is the manuscript technically sound, and do the data support the conclusions?

Reviewer #4: Yes

Reviewer #5: Yes

3. Has the statistical analysis been performed appropriately and rigorously? 

Reviewer #4: Yes

Reviewer #5: Yes

4. Have the authors made all data underlying the findings in their manuscript fully available?

Reviewer #4: Yes

Reviewer #5: Yes

5. Is the manuscript presented in an intelligible fashion and written in standard English?

Reviewer #4: Yes

Reviewer #5: Yes

6. Review Comments to the Author

Reviewer #4: Lastly, I would like to advise you to separate the conceptual differences between expectations and bias more carefully and reflect them in the manuscript.

Reviewer #5: (No Response)

7. PLOS authors have the option to publish the peer review history of their article (what does this mean?). If published, this will include your full peer review and any attached files.

Reviewer #4: No

Reviewer #5: No

---

## [Editor Report · Acceptance letter]

30 Jul 2021

PONE-D-20-16005R3 

Facial Femininity and Perceptions of Eating Disorders: A Reverse-Correlation Study 

Dear Dr. Balas:

I'm pleased to inform you that your manuscript has been deemed suitable for publication in PLOS ONE. Congratulations! Your manuscript is now with our production department. 

Kind regards, 

on behalf of

Dr. Kamila Czepczor-Bernat 

Guest Editor

PLOS ONE